# RGD-Coated Polymer Nanoworms for Enriching Cancer Stem Cells

**DOI:** 10.3390/cancers15010234

**Published:** 2022-12-30

**Authors:** Yushu Gu, Valentin Bobrin, Dayong Zhang, Bing Sun, Chun Ki Ng, Sung-Po R. Chen, Wenyi Gu, Michael J. Monteiro

**Affiliations:** 1Australian Institute for Bioengineering and Nanotechnology (AIBN), University of Queensland (UQ), St Lucia, Brisbane, QLD 4072, Australia; 2Department of Clinical Medicine, Zhejiang University City College, Hangzhou 310015, China

**Keywords:** RGD, nano-worm, CSCs, cancer stemness, colon cancer

## Abstract

**Simple Summary:**

Cancer stem cells are a small portion of tumor cells, and it is also hard to keep their stem features during isolation and culture, thus making the study of them very difficult. However, they play a critical role in tumor development, drug resistance, and metastasis. Therefore, an effective method to isolate them and maintain their stem feature is required. In this study, we explored a new method with our synthesized thermo-responsive polymer coated with triple peptides RGD to meet the requirement. We demonstrate that, with this system, we can promote cancer stem cell growth and sustain their stem nature. Furthermore, the new system does not need enzyme treatment to dissociate cells. This method can thus be widely used to isolate and culture cancer stem cells from primary tumor or cancer cell lines to facilitate the study of identifying therapeutic targets and screening drugs sensitive to cancer stem cells.

**Abstract:**

Cancer stem cells (CSCs) are primarily responsible for tumour drug resistance and metastasis; thus, targeting CSCs can be a promising approach to stop cancer recurrence. However, CSCs are small in numbers and readily differentiate into matured cancer cells, making the study of their biological features, including therapeutic targets, difficult. The use of three-dimensional (3D) culture systems to enrich CSCs has some limitations, including low sphere forming efficiency, enzymatic digestion that may damage surface proteins, and more importantly no means to sustain the stem properties. A responsive 3D polymer extracellular matrix (ECM) system coated with RGD was used to enrich CSCs, sustain stemness and avoid enzymatic dissociation. RGD was used as a targeting motif and a ligand to bind integrin receptors. We found that the system was able to increase sphere forming efficiency, promote the growth of spheric cells, and maintain stemness-associated properties compared to the current 3D culture. We showed that continuous culture for three generations of colon tumour spheroid led to the stem marker CD24 gradually increasing. Furthermore, the new system could enhance the cancer cell sphere forming ability for the difficult triple negative breast cancer cells, MBA-MD-231. The key stem gene expression for colon cancer also increased with the new system. Further studies indicated that the concentration of RGD, especially at high doses, could inhibit stemness. Taken together, our data demonstrate that our RGD-based ECM system can facilitate the enrichment of CSCs and now allow for the investigation of new therapeutic approaches for colorectal cancer or other cancers.

## 1. Introduction

Cancer stem cells (CSCs) consist of a small population of cells located within a tumour capable of self-renewal and differentiation to all cancer cell types, and is responsible for cancer imitation, development, and metastasis [1,2]. They are also involved in drug-resistance and tumour relapse [3,4]. The few CSCs within a solid tumour provides a significant challenge to isolate and identify them, in which there is currently no single or unique biomarker [5,6]. This, coupled with their rapid differentiation into mature cancer cells in a tumour microenvironment, makes it difficult to sustain and identify their stemness. Therefore, enriching CSCs would provide access to these and other inherent properties, and provide targeting strategies for both improved and lasting cancer treatments. 

Colorectal cancer (CRC) is one of the leading lethal cancers worldwide [7]. Approximately 1.4 million new diagnoses were reported in 2018 globally [8]. Conventional therapies and current cancer immunotherapy, such as immune checkpoint inhibitors and adoptive cellular therapy, are used to improve the survival rate of patients. However, drug resistance and tumour relapse are still major challenges for treatment strategies, and patients with CRC with therapeutic resistance have been closely linked to CSCs in the tumour [9]. The existence of CSCs has been demonstrated by the expression of several surface biomarkers previously identified and reported in CRC, including CD24, CD44, and CD133, which are commonly used to isolate CSCs from heterogeneous tumour cells [10,11]. In addition to identifying CSC populations, CD24 expression also indicates cell adhesion through extracellular matrix (ECM) activities [12]. Moreover, stem-related genes, including octamer-binding transcription factor-4 (OCT4), leucine-rich repeat-containing G-protein coupled receptor 5 (Lgr5), cellular Myelocytomatosis (c-Myc), src-homology 2 domain– containing phosphatase 2 (SHP2) and transforming growth factor β1 (TGF-β1), have also been proposed as means to identify colorectal CSCs [13]. Therefore, the investigation of these biomarkers is essential for understanding tumour heterogeneity, initiation, metastasis, and progression of CRC.

In vitro adhering (monolayer) cell culture is a common approach for studying tumour biology and developing therapeutic strategies (including new drug screening) against cancer. Two dimensional (2D) culture systems have many limitations as most tumour cells communicate with proteins in the ECM and with neighbouring cells in a 3D tumour microenvironment [14]. Cell–cell and cell–ECM communication that occurs within a 3D environment in vivo cannot be sufficiently replicated in 2D cell culture systems, thus limiting the effectiveness of these models when investigating cellular mechanisms and drug interactions. Subsequently, drugs that may appear effective in 2D culture systems in vitro may not translate to in vivo outcomes in patient treatments [15]. Many CSCs can be isolated from the whole cell populations through the specific cell surface markers (like CD24 and CD133 for colorectal cancer) using cell sorting methods such as flow cytometry-based fluorescence-activated cell sorting (FACS) and magnetic bead-based magnetic-activated cell sorting (MACS) [16]. Cell sorting methods remain a challenge due to the low percentages of CSCs in whole cell populations and the lack of universal CSC-specific cell markers for different cancer types [17]. For example, the expression of CD44+ and CD24-, regarded as CSC markers for breast cancer, cannot be used as accurate biomarkers to efficiently isolate CSCs from triple-negative breast cancer [18]. Cell sorting methods also have other limitations, including the high cost of equipment and cell injury from the high electrical current during the cell sorting.

Another common method used in many laboratories is tumour sphere culture for cancer cell growth in 3D with a small number of growth factors (serum-free) to promote the self-renewal ability of CSCs [19,20,21]. CSCs will grow into a tumour spheroid with some cells differentiating into non-CSCs. Typically sphere formation occurs at a slow rate due to the small amount of ECM in the system and the high concentration of BSA. In addition, the separation of CSCs from spheres requires treatment with trypsin and vigorous pipetting (or passing through syringe/needles) that may damage the CSC cell surface [22]. Therefore, a less destructive method is required to sustain and maintain stemness. 

The tripeptide motif Arg-Gly-Asp (RGD) within the ECM proteins, such as vitronectin and fibronectin, has been identified as a pivotal integrin ligand that supports cell adhesion [23]. RGD binds to eight of the most important integrin dimers out of the 24 receptors reported; for example, the integrins αvβ1and αvβ3 mediate cell proliferation whilst αvβ6 is associated with TGF-β1 activation and EMT [23,24]. In particular, the RGD motif is recognised with high specificity by the integrin αvβ3, which is known as a luminal progenitor cell marker for CSCs and plays a crucial role in tumour growth, metastasis, and angiogenesis [25]. A study by Hurt et al. also described how RGD inhibited the morphologic changes in prostate CSC spheres and maintains stemness even in the presence of 1% human serum [26]. It has been demonstrated that a thermoresponsive polymer sphere culture system coated with RGD can have application for enriching and maintaining stemness in CSCs [27]. 

A thermoresponsive nanoworm (NW) was produced using the temperature-directed morphology transformation (TDMT) method established by our research group [27,28]. Functionalization of the surface of the NWs with RGD is through the physical (i.e., without a covalent attachment) binding of GRGD chain-end functional poly N-isopropylacrylamide (PNIPAM-GRGD) chains. This allows for the surface density of RGD on the NW to be quantitatively controlled. By increasing the temperature above the lower critical solution temperature (LCST), the NW and PNIPAM-GRGD will bind to cancer cells to form tumour spheres (Figure 1) [29]. This thermoresponsive property of PNIPAM allows the cell aggregates to dissociate into single cells by simply lowering the temperature below LCST with gentle pipetting, and without the use of trypsin. Our method facilitates cell dissociation without membrane-disrupting enzymatic treatments and provides fully defined conditions for tumour sphere formation. RGD serves as a targeting motif as well as a trigger to activate integrin signalling pathways. Furthermore, the polymer contains side groups with comparable chemical structures (amino acid, lucine and valine) allowing for biocompatibility with different cell lines [29]. Using this system, we successfully increased the cell numbers and sustained the stemness of human embryonic stem cells [27,29]. In this study, the PNIPAM-RGD binds to cancer cells and then the added NWs form an ECM to promote cancer stemness. We found that activation of the RGD signal pathways and ECM both contribute to cancer stem cell proliferation and sustain their stemness, indicating that our system can be used to enrich CSCs at least in colon cancer for in vitro study and drug screening with a possible application to CSCs of other cancers. 

## 2. Methods and Materials

### 2.1. Materials

Unless otherwise stated, all chemicals were used as received and as reported before [30]. The solvents used were of either HPLC or AR grade; these included dichloromethane (DCM, Aldrich AR grade), chloroform (Emsure, ACS, Castle Hill, NSW, Australia), acetone (ChemSupply, AR, Gillman, SA, Australia), petroleum spirit (BR 40–60 °C, Univar, AR, Rocklea, QLD, Australia), methanol (Emsure, ACS), toluene (Emsure, ACS), acetone (ChemSupply, AR), glacial acetic acid (Merck, EMPROVE ESSENTIAL Ph Eur, BP, JP, USP, E 260), diethyl ether (Emsure, ACS), ethyl acetate (ChemSupply, AR) and *N*,*N*-dimethylacetamide (Aldrich, >99%). Activated basic alumina (Aldrich, Castle Hill, NSW, Australia: Brockmann I, standard grade, ~150 mesh, 58 Å), silica gel (Aldrich, technical grade, 230−400 mesh, 60 Å), magnesium sulphate (anhydrous, Scharlau, extra pure, Gillman, SA, Australia), Milli-Q water (Biolab, SA, Australia, 18.2 MΩm), sodium dodecyl sulphate (SDS, Aldrich, 99%), *N*,*N*′-dicyclohexylcarbodiimide (DCC, Aldrich, 99%), 4-(dimethylamino)pyridine (DMAP, Merck, 99%), 1-butanethiol (Aldrich, 99%), propargyl alcohol (Aldrich, 99%), triethylamine (Aldrich, 99.5%), 1-adamantanemethanol (Aldrich, 99%), Aldrithiol™-2 (DPDS, Aldrich, 98%), 2-mercaptoethanol (Merck, >98%), lithium chloride (Aldrich, 99%), tripotassium phosphate (Aldrich, ≥98%), sodium hydrogen carbonate (Aldrich, 99.5%), hydrochloric acid (36%, Ajax, AR, Macquarie Park, NSW, Australia), carbon disulfide (Aldrich, >99.9%), 2-bromo-2-methylpropionic acid (Aldrich, 98%) and methyl-2-bromopropionate (MBP, Aldrich, 98%) were used as received. Styrene (STY, Aldrich, >99%) and *N*,*N*-dimethylacrylamide (DMA, Aldrich, >99%) were passed through a basic alumina column to remove inhibitor. *N*-isopropylacrylamide (NIPAM, Aldrich, 97%) was recrystallized from n-hexane/toluene (9/1, *v*/*v*), and azobisisobutyronitrile (AIBN, Riedel-de Haen, Gillman, SA, Australia) was recrystallized from methanol twice prior to use.

### 2.2. Synthesis of Nanoworms (NWs)

The synthesis of nano-worm was basically as we reported previously [30]. Briefly, in a Schlenk tube, MacroCTA4 (75.6 mg, 1.6 × 10^−5^ mol), MacroCTA1 (65.6 mg, 9.3 × 10^−6^ mol), Alk-MacroCTA2 (33.8 mg, 5.5 × 10^−6^ mol) and SDS (7.25 mg, 2.5 × 10^−5^ mol) were dissolved in cold Milli-Q water (3.25 mL). The mixture was deoxygenated by purging with Argon for 20 min. AIBN (0.47 mg, 2.8 × 10^−6^ mol) was dissolved in styrene (0.1689 g, 1.6 × 10^−3^ mol) and the solution injected into the MacroCTAs mixture, which was then purged with Argon for another 5 min in an ice bath before heating to 70 °C. The polymerization was stopped after 4 h by exposing the reaction to air at 70 °C. The latex (1 mL) at 70 °C was mixed with 20 μL toluene, cooled to 23 °C, and left to stand at 23 °C for 30 min. The mixture was then cooled to 12 °C, and left to stand at 12 °C for 23 h. The worm structure was confirmed by TEM, and then 1 mL of the worm solution was diluted by adding 5 mL of Milli-Q water for biology assays. 

### 2.3. Conjugated RGD to PNIPAM (PNIPAM-GRGD)

RGD was synthesized as Azide-GRGD by GenicBio Limited (Shanghai, China). The conjugation of GRGD to PNIPAM and nano-worm and the characterization of these nano-structures were basically as described before [27]. 

### 2.4. The 3-D Tumour Sphere Culture

The human colon cancer cell line HCT116 was selected as the primary model for testing the new polymer culture system. HCT116 cells, and the triple-negative breast cancer cell line MBA-MB-231, were purchased from the American Type Culture Collection and maintained in Dulbecco’s Modified Eagle’s Medium (DMEM, Gibco, Perth, WA, Australia) containing 1% penicillin/streptomycin (P/S, 100X, Invitrogen, Mulgrave, VIC, Australia), and 10% foetal bovine serum (FBS, Gibco, Australia) in 5 mL flasks under 5% CO_2_ at 37 °C. After the cell confluence reached 80%, they were trypsinised to separate into single cells. The cells were counted, and 2 × 10^4^ cells were then suspended in DMEM/F12 (18) (1:1) (Gibco) supplemented with 5 μg/mL insulin, 1% B27 supplement (Gibco, Hackett Drive Crawley, WA, Australia), EGF (20 ng/mL; Sigma, Castle Hill, NSW, Australia), 2 grams of bovine serum albumin (BSA, BioReagent, SA, Australia), and 1% Penicillin/Streptomycin. This suspension system was used as the control group in the study with this medium referred to as the sphere culture medium (SCM). Both cell lines were regularly tested for mycoplasma contamination and only early passage generations, no later than 15 passages, were used in the sphere culture. 

### 2.5. New Sphere Culture System with PNIPAM-GRGD and NW System

The polymer PNIPAM-GRGD and NWs used in this study were similar to the system we previously published for human embryonic stem cells [27,29]. However, the P-fibronectin was replaced by PNIPAM-GRGD in this study. The HCT116 or MBA-MB-231 cells (5000 cells/mL unless indicated otherwise) were detached using 1X trypsin-EDTA when the cell confluence reached 80%. Cells were then suspended in SCM and plated into ultra-low attachment plates (Corning, Silverwater, NSW, Australia) with varying concentrations of the polymers added. The SCM temperature was set below the LCST prior to adding the polymers. In addition, for the PNIPAM-GRGD and NW system, PNIPAM-GRGD was added to the cells for 1 hour below the LCST prior to addition of NWs. After mixing, cells were cultured for a total of 7 days at 37 °C under 5% CO_2_ before the sphere culture was ready for enzyme-free passaging and further analysis. 

### 2.6. Enzyme-Free Cell Passaging

Enzyme-free cell passaging was performed by lowering the temperature of the HCT116 spheres to below LCST for 1 hour, enabling polymer aggregates to disappear. Spheres were harvested by centrifugation for five minutes at 100× *g*. Supernatants were discarded and 1 mL cold sterile 1X phosphate-buffered saline (PBS) was added to the pelleted spheres to maintain the temperature under LCST. A small portion of spheres for each nanoparticle condition and control group was taken and digested into single cells with 1x trypsin–EDTA for total spherical cell counting using haemocytometer. The remaining cells treated with nanoparticles were pipetted gently for 100 times to break the spheres into single cells. For the sub-culture, 5000 cells/mL cells were then suspended into fresh SCM. New nanoparticles were added to the cell cultures with the same concentrations as described above. After 4 days of incubation, enzyme-free passaging from the 1st was transferred to the 2nd generation culture, and the 2nd was similarly transferred to the 3rd generation culture.

### 2.7. Analysis of Sphere Forming Efficiency (SFE)

Spheres were harvested by centrifugation at 100× *g* for 5 min and then resuspended gently into 2 mL SCM. Spheres (100 μL/well) were then evenly transferred to into wells in an ultralow attachment 96-well plate for imaging under the light microscopy. Spheres with different diameters greater than 50 μm were imaged and scored. Sphere forming efficiency (SFE) was calculated by the scored number of spheres divided by the total number of cells seeded in each culture, that is: the SFE = total sphere numbers/total cell seeded in the culture × 100%. 

### 2.8. Flow Cytometry Analysis of Surface Markers

The spheres for FACS analysis were disassociated with 1X trypsin–EDTA and harvested by centrifugation at 300× *g* for 10 min. Supernatants were removed and 2 mL 1X PBS were used to resuspend the cell pellets and passed through a cell strainer (40 µM, BD, North Ryde, NSW, Australia) for cell counting. Cell counting ensured 0.5–1× 10^6^ cells were collected for each tube in 100 µL volume. The antibody conjugates were then added. After incubation at room temperature in darkness for 1 h, the tubes were centrifuged at 300× *g* for 10 min to remove the supernatants. The pellets were then washed twice with 1 mL 1X PBS, resuspended in 100 μL 1X PBS and fixed with 4% paraffinformaldehyde (PFA) for flow cytometry analysis. The antibodies and isotype controls used in FACS analysis included mouse FITC-conjugated CD24 monoclonal antibody M1/69 (1:200 dilution); mouse FITC-conjugated IgG2b, isotype control (1:500 dilution); mouse PE-conjugated CD44 monoclonal antibody IM7 (1:800 dilution); mouse PE-conjugated CD133 (Prominin-1) monoclonal antibody 13A4 (l:100 dilution); and mouse PE-conjugated IgG1, kappa isotype (1:500 dilution). The samples were acquired on a CytoFLEX Flow Cytometer (Beckman Coulter, Gladesville, NSW, Australia) and analysed using CytExpert (Beckman Coulter). 

### 2.9. Quantitative Real-Time Polymerase Chain Reaction (qRT-PCR)

Total RNA was prepared from the harvested cell pellets using the TRIzol reagent (Invitrogen, Mulgrave, VIC, Australia) according to the manufacturer’s instructions. NanoDrop (Thermo Scientific, Scoresby, VIC, Australia) was used to analyse mRNA quality in the samples and 10 μl of RNA sample were mixed with 20 μL system high-capacity cDNA reverse transcription kit in PCR tubes according to the manufacturer’s protocol. Reverse transcription was performed with a PCR thermal cycler (Thermo Fisher Scientific, Scoresby, VIC, Australia) at 25 °C for 10 min, 37 °C for 120 min, and 85 °C for 5 min. The resulting cDNA was then diluted from 20 μL to 200 μL with sterile water before qRT-PCR. The qRT-PCR was performed with SYBR^®^ Green PCR master mix (Applied biosystems, Mulgrave, VIC, Australia) to measure transcription levels of stemness-related genes qRT-PCR was performed in the CFX96 Real-Time PCR Detection System (Bio-Rad, Gladesville, NSW, Australia) under a cycling protocol of 95 °C for 5 min, followed by 40 cycles of 95 °C for 15 s and 40 cycles of 60 °C for 1 min. Data analysis was performed using Bio-Rad CFX Manager Software using the 2^−ΔΔCt^ method. GAPDH was used as the internal control to normalise the gene expression data. Transcription of each gene was compared to the transcription of parental cells and presented as fold changes. The sequences of the specific primers were (purchased from Integrated DNA Technologies, San Diego, CA, USA): GAPDH: Forward: 5′-CTTTTGCGTCGCCAG-3′, Reverse: 5′-TTGATGGCAACAATATCCAC-3′; CD133: Forward: 5′-CACCAAGCACAGAGGGTCAT-3′, Reverse: 5′-CACTACCAAGGACAAGGCGT-3′; OCT4: Forward: 5′-CAAAGCAGAAACCCTCGTGC -3′, Reverse: 5′- AACCACACTCGGACCACATC-3′; SHP2: Forward: 5′-AGAGCCACCCTGGAGATTTT-3′, Reverse: 5′-CTCCTCCACCAACGTCGTAT-3′; TGF-β1: Forward: 5′-CAACAATTCCTGGCGATACC-3′, Reverse: 5′-GAACCCGTTGATGTCCACTT-3′; Lgr5: Forward 5′- GATGTTGCTCAGGGTGGACT-3′, Reverse: 5′- GGGAGCAGCTGACTGATGTT-3′; c-Myc: Forward: 5′-TACAACACCCGAGCAAGGAC-3′, Reverse: 5′-TCCTCCTCGTCGCAGTAGAA-3′; NANOG: Forward 5′-GGTGGAGTATGGTTGGAGCC-3′, Reverse: 5′-AATATTAGCCGGGCGAGGTG-3′; Integrin α2: Forward: 5′-CCTACAATGTTGGTCTCCCAGA-3′, Reverse: 5′-AGTAACCAGTTGCCTTTTGGATT-3′; Integrin β1: Forward: 5′-CCTACTTCTGCACGATGTGATG-3′, Reverse: 5′-CCTTTGCTACGGTTGGTTACATT-3′. 

### 2.10. Limiting Dilution Assay (LDA) for HCT116

The LDA was used to determine the CSC frequency of colorectal cancer cells enriched with our new culture system in comparison to the control group of cells maintained only in SCM. The colorectal cancer HCT116 cells were cultured, harvested, and counted using a haemocytometer. The cells were serially diluted to prepare the final cell numbers needed (1 cell/well, 5 cells/well, 10 cells/well, and 20 cells/well in 100 µm volume) for seeding in ultra-low attachment 96-well cell culture plates (with two biological repeats). The wells with cells were checked and marked the next day (especially for 1 cell/well). All cells were grown at 37 °C for 7 days before a microscope examination to determine cell sphere numbers in each well. The data was recorded and put into the extreme limiting dilution analysis (ELDA) software to determine the frequency of sphere formation. 

### 2.11. CFSE Staining for HCT116 Cells and Cell Sorting

HCT116 cells were cultured in T25 flasks as monolayer cells. When they reached 80% confluent, they were harvested by trypsin and washed with 1X PBS. The cells were then stained with CFSE according to the manufacturer’s menu (BD Biosciences, North Ryde, NSW, Australia) and then cultured for a further 3 days before sorting for fast-dividing cells. The fast-, median-, and slow-dividing cells were classified by cells at the lowest 15%, median 15%, and highest 15% of CFSE intensity in a histogram display (Refer Appendix A). The fast-dividing cells were set up for sphere culture. 

### 2.12. Statistical Analysis

All data are expressed as mean ± standard deviation (SD) with 3 biological repeats. Statistical analysis was conducted using GraphPad Prism software 8.0.2 (GraphPad Software, Inc., San Diego, CA, USA). Statistical comparison was performed using one-way ANOVA and Turkey multiple comparisons or a two-tailed Student *t*-test. A *p* value < 0.05 was deemed as a significant difference. 

## 3. Results

### 3.1. Synthesis of Polymeric Nanoparticles and Working Scheme

The NWs are then given the ability to provide comparable binding sites to those on the ECM when the temperature is raised above the lower critical solution temperature (LCST) in the presence of PNIPAM-GRGD, a synthetic peptide with an RGD integrin recognition sequence coupled to a polymer chain. The thermoresponsive PNIPAM has the ability to undergo a phase transition from globule (water-insoluble) to coil (water-soluble) when the temperature is lowered below the LCST (Figure 1) [27,29].

### 3.2. The Effect of RGD and RGD-ECM on Sphere Formation and Growth

It is well known that RGD can modulate cell proliferation, migration, and apoptosis via interacting with integrin receptors and activating related signalling pathways [23]. It is also reported that the RGD within vitronectin or fibronectin can promote cell growth and sustain the stemness of embryonic stem cells [29]. We synthesised PNIPAM conjugated directly with RGD (PNIPAM-GRGD; P-GRGD, the first G was used to conjugate with polymers) to form a tumour spheroid culture system of colon cancer cells HCT116, which is a well-established CSC culture model in our Lab [11,31]. We first compared different doses of free GRGD, P-GRGD alone, and P-GRGD+NW treatment on sphere forming efficiency (SFE) in HCT116 cells after a seven-day sphere culture. The GRGD concentration for P-GRGD was the same as free GRGD. The results showed that free GRGD treatment excreted a dose-dependent increase of SFE within 5 µg/mL doses (Figure 2A). The treatment with P-GRGD alone also showed the same trend within the same GRGD dose range, suggesting RGD alone or with PNIPAM can increase SFE in a dose-dependent manner (Figure 2A). The addition of nanoworm (at 5 µg/mL) in the culture with different concentrations of P-GRGD showed a dose-dependent increase of SFE at doses between 0.1–0.5 µg/mL, while at higher doses (1.0–5.0 µg/mL) it exhibited a dose-dependent decrease of SFE (Figure 2A). The highest SFE with P-GRGD+NW system was found at a P-GRGD concentration of 0.5 µg/mL and was higher than both free GRGD or P-GRGD alone at 5 µg/mL (increased about 1.35 times), suggesting the ECM formed by the nanoworms sensitised the CSC response to RGD activation. These results also suggest that 0.5 µg/mL P-GRGD and 5.0 µg/mL nanoworm is the optimised dose. More importantly, this comparison study reveals that the cancer cells respond to both RGD and its ECM treatment, and the polymer matrix changed the sensitivity of cancer cells responding to an RGD signal. At higher P-GRGD doses, cancer cells exhibited an inhibitory response to RGD; this warrants further investigation on its therapeutic values. 

We next investigated the influence of P-GRGD dose at 0.5 µg/mL NW on tumour sphere formation and growth. The total sphere number (Figure 2B and Appendix A) significantly increased from a P-GRGD dose of 0.5 to 2.5 µg/mL compared to the control (i.e., without any treatment). At a dose of 5.0 µg/mL, the sphere numbers significantly decreased while the size increased due to a strong aggregation (Figure 2B). Further analysis of the size distribution of the spheres showed an influence of P-GRGD dose (Figure 2C). At a dose of 0.5 µg/mL, the sphere size increase to 200-300 µm, whereas at 1.0 µg/mL the spheres sizes were between 100–200 and <100 µm. The higher doses of 2.5 and 5.0 µg/mL mainly increased the spheres to sizes between 200–300 µm and larger. The data suggest that, at higher P-GRGD concentrations, it was larger with a lower number of spheres found. To describe the effect of the P-GRGD+NW system on individual spherical cell growth in more depth, we calculate the cell number per sphere according to the method described in Appendix A. The results showed that, for the total spherical cell numbers in all spheres, the 0.5 µg/mL P-GRGD+NW condition was much higher than any other condition used (Figure 2D). At the dose from 1.0 to 2.5 µg/mL, the spherical cell numbers were still significantly higher than the control and exhibited a dose-dependent increase due to a greater number of larger spheres. The data demonstrate that the P-GRGD+NW treatment can increase SFE, spheroid numbers and spherical cell numbers at an optimised dose of 0.5 µg/mL P-GRGD (at 5 µg/mL NW). 

The SFE is an important indicator of the self-renewal ability of cancer cells as a marker for stemness [32,33]. The increased SFE demonstrates the elevated stemness of HCT116 cells in the presence of the P-GRGD+NW system, and together with the increased spherical cell numbers suggests that our system has promoted spherical cell growth. This increase in both sphere formation and cell number (growth) represents a good reflection of CSCs expansion in vitro [19,31].

### 3.3. The Effect of RGD and RGD-ECM on Surface Markers of Colon CSCs

We next analysed the spherical cells with some common colon CSC markers, including CD24, CD44, and CD133 [11,22,34,35], to identify the correlation between sphere size and CSC formation. First, we carried out assays to compare different doses of free GRGD, P-GRGD alone, and the P-GRGD+NW treatment on marker expression of spherical cells after seven days sphere culture of HCT116 cells. The results showed that all three CSC markers (Figure 3A) followed the same trend as for SFE in Figure 2A. That is, the treatment of free GRGD and P-GRGD alone showed the dose-dependent increases of all surface marker expressions, whereas the P-GRGD+NW treatment showed a maximum for all three markers at 0.5 µg/mL P-GRGD. Taken together, the data in Figure 2 and Figure 3 indicate that P-GRGD+NW can increase cancer stemness, producing more sensitive CSCs that selectively responded to an RGD signal. This also implies that at high concentrations of P-GRGD+NW can inhibit cancer stem cell proliferation and stemness expression, which warrants further investigation for its therapeutic potentials. 

In previous studies, RGD was considered as only a targeting motif on a scaffold or ECM [36,37,38]. In this study, we showed that the combination of P-GRGD and NW could alter CSC growth and stemness expression by activating the different integrin pathways. To understand this, we carried out studies on the integrin alpha and beta receptor, in which our real-time RT-PCR results showed that mRNAs of integrin alpha and beta receptor expression also increased after treatment with P-GRGD+NW (Appendix A). This supports that RGD can also activate different downstream signalling pathways [23], leading to cell survival, proliferation, and even apoptosis. We then conducted a detailed study of stem marker expression about the optimised condition of 0.5 µg/mL P-GRGD. The results showed that, remarkably, all the biomarker expressions increased after the treatment with the P-GRGD+NW system, with the percentage of positive cells increasing from 36.0% to 86.6% for CD24, 78.3% to 94.3% for CD44, and 61.9% to 81.7% for CD133 (Figure 3B) together with the mean fluorescence intensity (MFI, Figure 3C). Among these markers, CD24 expression increased more significantly, suggesting that this stem marker may be more sensitive than the other two markers. 

### 3.4. The Effect of RGD and RGD-ECM on Stem Gene Expressions

The stem gene profile was examined after treatment with different doses of free GRGD, P-GRGD, and P-GRGD+NW at day seven of the sphere culture of HCT116 cells. However, unlike the SFE and surface biomarker data, the gene profile did not show the clear dose-dependent trend found for our P-GRGD+NW system (Figure 4A). For the three typical stem genes, including C-Myc, CD133, and SHP2, C-Myc and CD133 exhibited a similar pattern to SFE (see Figure 2) and surface stem markers (see Figure 3) after P-GRGD+NW treatment. We also did not observe the dose-dependent increase after either GRGD or P-GRGD treatment. Only SHP2 exhibited an almost dose-dependent increase of expression after P-GRGD+NW treatment (Figure 4A). Other feature stem gene expressions, including OCT4, NANOG, Lgr5, and TGF-β1, did not show either a dose-dependent increase or decrease (Appendix A), suggesting that the stem gene expression may not be at the same expression pace as those surface biomarkers. To further understand the details of the stem gene expression, we measured the gene expressions of signature stem and self-renewal genes (including OCT4, NANOG, C-Myc, CD133, Lgr5, TGF-β1, and SHP2) of colon cancer in the treated cells with the optimised conditions (Figure 4B). Almost all gene expressions (except OCT4) increased in the culture with P-GRGD+NW system. Among them, C-Myc, CD133, and SHP2 reached statistical significance (*p* < 0.05), while the others did not. For these three major gene profiles, after P-GRGD+NW treatment at 0.5 µg/mL they all increased in comparison to the control, consistent with the results in Figure 4A. Taken together, the collective data indicate that the P-GRGD+NW system can increase sphere formation and spherical cell growth while promoting stemness in terms of signature stem surface and gene markers.

### 3.5. The P-GRGD+NW System on Sustaining and Promoting Cancer Stemness

To further explore whether the system could sustain stemness, we examined if continuous culturing of the tumour spheroid with our P-GRGD+NW system will sustain and maintain the stemness using CD24 as the representative marker. We monitored three continuous generations of sphere culture. As shown in Figure 5A, the CD24 expression increased with the generation number, suggesting that the P-GRGD+NW system can further increase stemness and maintain the CD24 surface biomarkers over a continuous treatment period. CD24 is indeed an important biomarker to maintain stemness at least in colon cancer. 

We carried out three more experiments to further demonstrate the ability of our system to promote stemness in a sphere culture. The first was the limiting dilution assay (LDA). Here, we diluted the cells to seed one cancer cell in each well at 100 µL volume in 96-well plates and evaluated if a single cell could form a tumour spheroid with the P-GRGD+NW system. Cell numbers in each well were viewed and recorded at day one after seeding and spheroid numbers were recorded at day seven. The results revealed a significant difference in the SFE frequency per cell between the control and the P-GRGD+NW system (Figure 5B). In the control, the average number of cells per spheroid was 8.50 with a confidence interval of 7.21–10.05, whilst the P-GRGD+NW group required a much lower number of cells of 5.61 with a confidence interval of 4.79–6.58. This result supports the high sphere-forming efficiency of individual cancer cells when cultured with the P-GRGD+NW system. 

In the second experiment, we sorted HCT116 cells stained with CFSE dye into three proliferation populations (i.e., fast, median and low) according to their MFI using flow cytometry (Appendix A). It was previously shown that fast-dividing cells had a much lower sphere formation potential than low-dividing glioblastoma initiating cells [39]. Our P-GRGD+NW system significantly increased the SFE (Figure 5C) and also the signature genes (including stem gene OCT4 and self-renewal gene TGF-β1) expressions (Figure 5D). The third assay tested if the system could promote a cancer line that normally is difficult to form spheres. We chose the triple negative breast cancer MDA-MB-231 cell line as a model, which is normally unable to form spheroids in routine sphere culture systems (i.e., Control). When using our P-GRGD+NW system, this cell line formed typical spheroids (Figure 5E). In summary, our P-GRGD+NW system facilitates both sphere-forming ability and the stemness of individual cancer cells. 

## 4. Discussion

Isolating CSCs from cancer cell lines or primary tumours and sustaining their stemness are critical steps for studying CSC properties and developing targeted therapies. To achieve this, in this study, we have synthesised P-GRGD and NW systems and shown that routine tumour sphere culture with P-GRGD+NW system can greatly increase the stemness of cancer cells and sustain their stemness, indicating a wide application of this system in the CSC study. Additionally, as a thermoresponsive system, the passage and expansion of CSCs are free of trypsin/EDTA treatment, thus effectively protecting CSC membranes from damage. 

There are not many studies on how RGD peptide affects CSCs. Our results are consistent with the recent study showing that CSCs respond differently to RGD doses in a hydrogel system [40]. The data supports that ECMs coated with RGD both modulate CSC proliferation and stemness. The stochastic distribution of RGD on the surface of the hydrogel system compared to the natural binding of RGD to the cell surface using the P-GRGD may influence the signalling transduction in colon cancer cells and thus moderate CSC proliferation and stemness. Some studies have reported the RGD activation of integrin pathways and promotion of cancer cell proliferation and metastasis [23,41], which is consistent with our current findings with GRGD or P-GRGD alone. Our data with both P-GRGD and NWs can not only promote but also inhibit sphere formation and cell growth depending on the RGD dose. The increase of cell proliferation and stemness is proposed to be due to the combination of an ECM and activation of RGD signalling pathway. This finding supports previous work in which RGD coated hydrogels enhanced cell aggregates and drug resistance formed on the hydrogel surface at low RGD functionality, while at high RGD functionality this effect was not as pronounced [40]. RGD was also used with Ketoprofen to target breast cancer stem-like cells for cancer therapy [36], indicating that RGD binds specifically to CSCs.

Our P-GRGD+NW system can promote and maintain stemness. This is in agreement with the role of other ECMs in the development of normal intestinal stem cells that are crucial in colon epithelium physiology [42] and also in the CSCs of various cancer types [43,44]. ECMs are an essential component of the tumor microenvironment during tumor development and progression via regulating cancer cell phenotypes. A study reported the use of ECM-based hydrogels for tailoring tumor organoids and highlighted the potential role of the ECM in the development of recapitulating malignant and invasive tumor organoids with enhanced capacity for in vitro representation of ECM-regulated tumor progression [45]. In our study, our P-GRGD+NW system has a thermoresponsive behaviour that enables us to isolate cells from the spheroid, resulting in the protection of the CSC surface markers of CSCs that are damaged when using trypsin. 

Among all stemness markers, the three surface ones chosen in this study for colon CSCs are well studied and accepted [16]. Among them, CD24 was shown to play an important role in clustering/connecting the colon CSCs and contribute to the metastatic potential of colon cancer [22]. CD24 also played a role in immune regulation as an inhibitory factor to mediate immune escape via interaction with Siglec-10 on macrophages in breast and ovarian cancers [46], suggesting its critical biological function in cancer development. CD133 and CD44 are also adherent molecules and are shown to be important stem biomarkers for colon [35,47] and other cancers [48,49]. As a good example, CD44 has been used as a critical cancer cell/CSC therapeutic target for targeted drug delivery and cancer therapy [50,51,52]. 

This inconsistent disparity between surface marker and gene expression suggests that the two profiles were not synchronised or not at the same expression pace. This is especially the case for CD133 (Figure 3 vs. Figure 4), in which the gene expression is normally earlier and short lived compared to protein expression. The other stem genes are associated with different pathways and may express over different time courses after the treatment. Further investigation of stemness gene profile over a time course is needed together with other stemness related proteins and signal pathways to identify if P-GRGD+NW regulates the surface marker rather at a greater rate than other stem gene profiles. Real-time RT-PCR is a sensitive method and could distinguish the variation between individual runs. Another possible reason is that the P-GRGD+NW system may have a greater influence on the cellular surface markers rather than other stem gene profiles as surface markers are directly relevant to sphere formation and the RGD signal. These results suggest that P-GRGD+NW treatment increases but does not decrease the major stemness gene expression even at high RGD doses. Moreover, the results also indicate that P-GRGD+NW is more sensitive and heavily influenced by the stemness of surface markers of CSCs rather than stem related gene profiles. In effect, the P-GRGD+NW system has a longer-term effect on promoting the stem gene profile for CSCs. The stem gene results found here suggest that P-GRGD+NW treatment can increase the major stem gene profile of colon CSCs. 

## 5. Conclusions

In this study, we have demonstrated that the RGD conjugated nanoworm system can not only increase cancer stemness (in terms of sphere formation, stem surface marker expression, and stem marker gene expression) and promote their growth but also sustain the cancer stemness, especially the surface markers. Additionally, this system is thermoresponsive and can expand CSC without the use of trypsin/EDTA treatment, suggesting that it will be a useful system to cultivate CSCs in vitro for colon cancer and other cancers as well. Another important finding is that we demonstrate that the RGD conjugated ECM system has the ability to inhibit CSC growth in a dose-dependent manner, especially at high dose levels. This warrants further investigation as a potential therapy approach targeting CSCs.

## Figures and Tables

**Figure 1 cancers-15-00234-f001:**
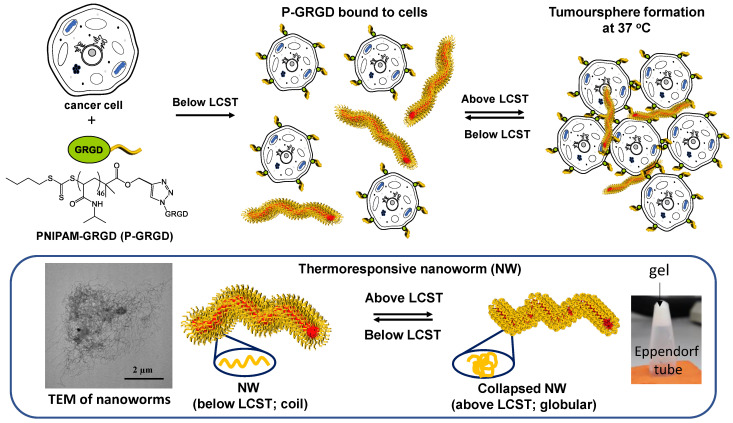
Schematic illustration how the RGD polymeric matrix work. PNIPAM-GRGD (P-GRGD) below LCST binds to the cancer cell surface, and after the addition of nanoworms (NWs), aids in the formation of cancer cell spheres. This process promotes the stemness and proliferation required for stem cell development. Lowering the temperature below the LCST allows for the dissociation of cells to be used in next generation cultures. The right picture of the bottom panel is the TEM image of nanoworm, and the left one is the gel image of the nanoworm.

**Figure 2 cancers-15-00234-f002:**
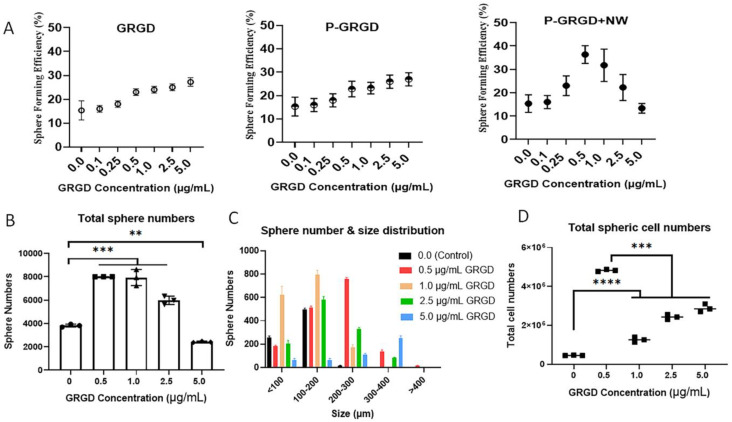
Effect of RGD dose on sphere formation and spheric cell growth. (**A**) SFE after treatment with free GRGD, PNIPAM-GRGD (P-GRGD) at the same doses of RGD as free GRGD, and P-GRGD+NW (5.0 µg/mL nano-worm). (**B**) total sphere number after cell culture for P-GRGD+NW system. (**C**) sphere size distribution at culture using the P-GRGD+NW system. (**D**) total spherical cell numbers in the spheres using the P-GRGD+NW system. Notes: **: *p* < 0.01; ***: *p* < 0.001; ****: *p* < 0.0001 (*n* = 3). The amount of NW used in these experiments was 5.0 µg/mL.

**Figure 3 cancers-15-00234-f003:**
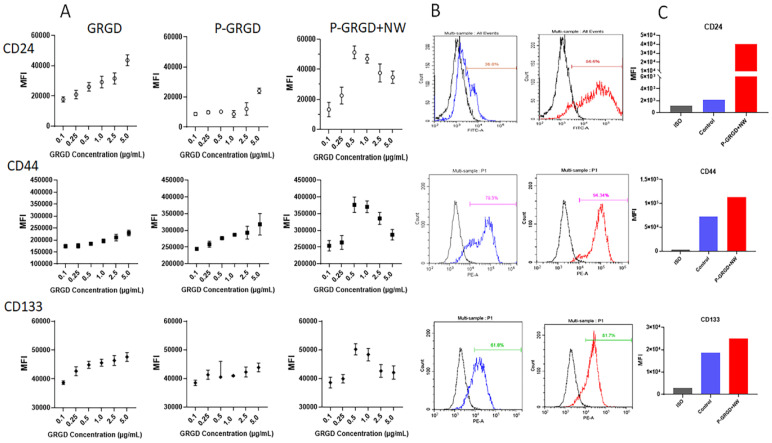
Influence of P-GRGD+NW treatment on the expression of surface stem markers in colon cancer cells. (**A**) flow cytometry analysis of dose-dependent expression for CSC biomarkers CD24, CD44, and CD133 after treatment with GRGD, P-GRGD, and P-GRGD+NW. (**B**) flow cytometry analysis of colon CSC surface marker expressions for CD24, CD44, and CD133 after 0.5 µg/mL P-GRGD plus 5 µg/mL NW treatment. (**C**) mean fluorescent intensity (MFI) of (**B**).

**Figure 4 cancers-15-00234-f004:**
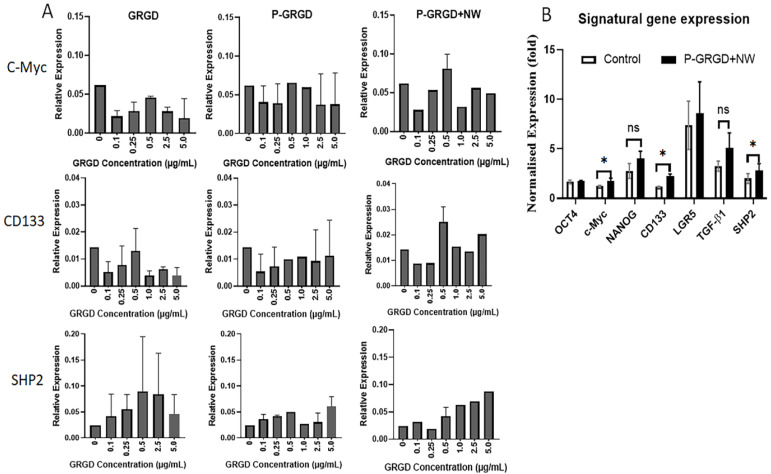
Influence of P-GRGD+NW treatment on expression of stem genes in colon cancer cells. (**A**) the real-time RT-PCR measurement of the featural cancer stem gene expressions, including C-Myc, CD133 and SHP2 after seven days sphere culture of HCT116 and treatment with different doses of GRGD, P-GRGD, and P-GRGD+NW. (**B**) some important stem gene expressions by qPCR measurement after seven days treatment with the optimised condition of 0.5 µg/mL P-GRGD plus 5 µg/mL NW in HCT116 cells. Notes: ns: *p* > 0.05; *: *p* < 0.05.

**Figure 5 cancers-15-00234-f005:**
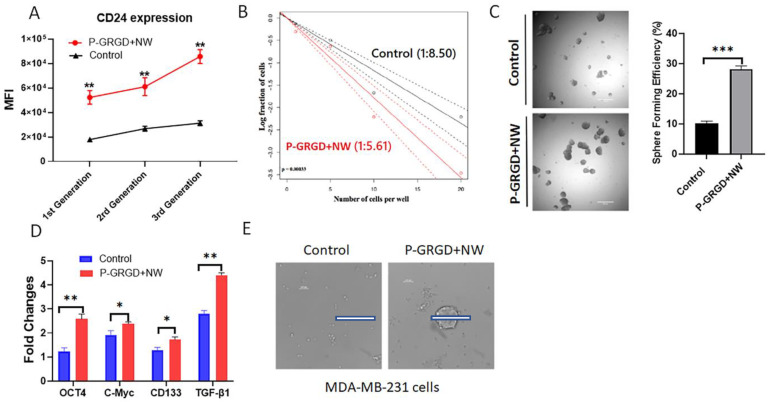
The RGD-ECM system more broadly affects cancer stemness. (**A**) the CD24 expression of CSCs after 3 continuous generations of sphere culture. (**B**) the limiting dilution assay shows that comparing to the control that needs average 8.5 cells to form a sphere, PG+W only needs an average of 5.6 cells to form a sphere. (**C**) the sorted fast-dividing HCT116 cells exhibited a much better sphere forming efficiency after being cultured with PG+W. (**D**) qRT-PCR results of fast-dividing HCT116 cells showing a better ability of expressing signature stem genes after treatment with PG+W. (**E**) MDA-MB-231 breast cancer cells that normally do not form a sphere have formed some typical spheres after being cultured with PG+W. Notes: *: *p* < 0.05; **: *p* < 0.01; ***: *p* < 0.001 (*n* = 3). The scale bars in (**C**,**E**) indicate 500 µm.

## Data Availability

Data is contained within the article can be further discussed by contacting the corresponding author.

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
