# Peer review of "RGD-Coated Polymer Nanoworms for Enriching Cancer Stem Cells"

_cancers, 2022, doi:10.3390/cancers15010234_

Round 1
Reviewer 1 Report
1. Please find a readable way to display the concept of RGD. It’s unclear if you used the RGD in the title and abstract.
2. Add TEM, SEM, etc. in detail to visualize the characterizations of polymer nanoworms in this manuscript.
3. What are the main purposes of enriching cancer stem cells in this study?
4. What are the differences between the tripeptide motif Arg-Gly-Asp (RGD) and commercially available polymer to support cell adhesion? Do you have contrast experiments?
5. Please provide real images of the 3D polymer extracellular matrix (ECM) system coated with RGD.
6. Add more analysis and explanation about the surface markers.
7. I think it’s excellent research. I suggest the authors provide a detailed analysis of the novelties in this research from the published papers. I wonder if this research first used RGD in the 3D polymer extracellular matrix system for cancer stem cell proliferation.
Author Response
Responses to reviewer’s comments
Review 1:
- Please find a readable way to display the concept of RGD. It’s unclear if you used the RGD in the title and abstract.
Answer: we have added a sentence in Abstract to explain why use RGD.
- Add TEM, SEM, etc. in detail to visualize the characterizations of polymer nanoworms in this manuscript.
Answer. We have added the TEM of the polymer nanoworms in Figure 1.
- What are the main purposes of enriching cancer stem cells in this study?
Answer: As we stated in the Introduction and Discussion, enriching cancer stem cells is mainly for in vitro studies of these cells, including identifying therapeutic targets and screening drugs sensitive to them.
- What are the differences between the tripeptide motif Arg-Gly-Asp (RGD) and commercially available polymer to support cell adhesion? Do you have contrast experiments?
Answer: we used RGD in our system for two purposes: one is as a cell targeting motif to binding to cell surface; another is that it can bind to integrin receptor to activate RGD/FAK pathways and then modulate cell proliferation and differentiation. In addition, our polymer has a unique thermoresponsive feature, that facilitate the cell separation, commercially available polymers usually don’t have such advantages and we thus did not compare them with our polymer.
- Please provide real images of the 3D polymer extracellular matrix (ECM) system coated with RGD.
Answer. We have also included a photo of the gel formation above the LCST in Figure 1.
- Add more analysis and explanation about the surface markers.
Answer: we have discussed and explained the surface marker in detail in Discussion (Page 13-14). They are well-studied and commonly used stemness markers for colon cancer and other cancers.
- I think it’s excellent research. I suggest the authors provide a detailed analysis of the novelties in this research from the published papers. I wonder if this research first used RGD in the 3D polymer extracellular matrix system for cancer stem cell proliferation.
Answer: thank you very much for the positive comments. There are a few similar studies reported recently but the polymer used are different from us, from this aspect, we are the first to use RGD-ECM for cancer stem cells.
Reviewer 2 Report
In previous work, the authors demonstrated a thermoresponsive polymer sphere culture system coated with RGD could be applied to hESC culturing (doi:10.1002/pola.29342). In the present study, Gu et al. aimed to induce tumor stem cells with this system. The results showed that the RGD conjugated nanoworm system can promote and sustain stemness of cancer cells (especially sphere formation and stem surface marker expression). Additionally, this system is thermoresponsive and can expand CSC without trypsin/EDTA treatment, keeping the cell surface intact. However, more evidence is needed to confirm the feasibility and value of the system.
Major comments:
1. As has been reported that RGD could inhibit the morphologic changes in CSC spheres and maintain stemness of spheres (doi: 10.1002/stem.271), the key point of this article is whether the RGD-coated thermoresponsive polymer sphere culture system performs better than regular RGD sphere culture system. To test this hypothesis, the authors should calculate the values of P-GRGD+NW vs GRGD in all experiments of this work.
2. In enzyme-free cell passaging, supernatants were discarded and 1ml cold sterile 1X phosphate-buffered saline (PBS) was added to the pelleted spheres to maintain the temperature under LCST. What is the temperature of the cold PBS? The authors should be aware that profound hypothermia enhances cell adhesion by increasing the stability of E-cadherins; mild hypothermia increases stem cell survival by reducing oxidative stress and prevents apoptosis via overexpressing anti-apoptotic heat shock proteins and mild-hypothermia also promotes cell proliferation by affecting gene expression in several ways (DOI: 10.2174/1574888X16666201229124842). All control groups in this work must be subjected to the same cold shock treatment.
Minor comments:
1. The pan OCT4 primer cannot indicate the pluripotency of cancer cells (Fig5D), please use OCT4A primer (DOI: 10.1038/s41419-018-0606-x) for qRT-PCR.
2. Immunoblotting of stemness genes is needed to confirm the results obtained with qRT-PCR.
Author Response
Responses to reviewer’s comments
Review 2:
In previous work, the authors demonstrated a thermoresponsive polymer sphere culture system coated with RGD could be applied to hESC culturing (doi:10.1002/pola.29342). In the present study, Gu et al. aimed to induce tumor stem cells with this system. The results showed that the RGD conjugated nanoworm system can promote and sustain stemness of cancer cells (especially sphere formation and stem surface marker expression). Additionally, this system is thermoresponsive and can expand CSC without trypsin/EDTA treatment, keeping the cell surface intact. However, more evidence is needed to confirm the feasibility and value of the system.
Answer: Thanks for the comments. We agree that further validation studies of the system are needed. However, to achieve this an initial in vitro study is essential to test and confirm the system working, which was the major goal of the current study.
Major comments:
- As has been reported that RGD could inhibit the morphologic changes in CSC spheres and maintain stemness of spheres (doi: 10.1002/stem.271), the key point of this article is whether the RGD-coated thermoresponsive polymer sphere culture system performs better than regular RGD sphere culture system. To test this hypothesis, the authors should calculate the values of P-GRGD+NW vs GRGD in all experiments of this work.
Answer: Thank you for the comment. We have calculated the increase time for suitable comparison study and added to the text.
- In enzyme-free cell passaging, supernatants were discarded and 1ml cold sterile 1X phosphate-buffered saline (PBS) was added to the pelleted spheres to maintain the temperature under LCST. What is the temperature of the cold PBS? The authors should be aware that profound hypothermia enhances cell adhesion by increasing the stability of E-cadherins; mild hypothermia increases stem cell survival by reducing oxidative stress and prevents apoptosis via overexpressing anti-apoptotic heat shock proteins and mild-hypothermia also promotes cell proliferation by affecting gene expression in several ways (DOI: 10.2174/1574888X16666201229124842). All control groups in this work must be subjected to the same cold shock treatment.
Answer: thanks for the comment, the cold PBS means the solution kept in 2-8oC fridge. When we use, we take it out from the fridge and sterilize with 70% ethanol and use it in the cell culture hood. All the group were treated same.
Minor comments:
- The pan OCT4 primer cannot indicate the pluripotency of cancer cells (Fig5D), please use OCT4A primer (DOI: 10.1038/s41419-018-0606-x) for qRT-PCR.
Answer: yes, we checked and blasted the primers we used for OCT4 they can amplify OCT4A isoform.
- Immunoblotting of stemness genes is needed to confirm the results obtained with qRT-PCR.
Answer: we agree with you and have thought about this but at day 7 of sphere culture, the effect of RGD on stemness genes and protein may be over. So, to investigate this, we may need carry out a series time course study on both gene and protein expression to really work out their relations. A risk of this study is that even we carry out the study not all genes will be regulated by RGD-Integrin or RDG-ECM activation, which will make the study complicated.
Round 2
Reviewer 2 Report
The authors did not conduct the experiments I suggested, but overall the manuscript is acceptable for publication after minor language polishing.
Author Response
Many thanks for the suggestion, we have checked the document and did find some spelling/expression mistakes, which are now corrected in "Track change".